# Can Homecare Chronic Respiratory Disease Patients with Home Oxygen Treatment (HOT) in Southern Okinawa, Japan Be Evacuated Ahead of the Next Anticipated Tsunami?

**DOI:** 10.3390/ijerph19095647

**Published:** 2022-05-06

**Authors:** Hiroshi Sekiguchi, Rie Takeuchi, Yoko Sato, Tsuyoshi Matsumoto, Jun Kobayashi, Takehiro Umemura

**Affiliations:** 1Department of Nursing for Home and Chronic Care, Graduate School of Health Sciences, University of the Ryukyus, Okinawa 903-0215, Japan; 2Department of Global Health, Graduate School of Health Sciences, University of the Ryukyus, Okinawa 903-0215, Japan; rietake@med.u-ryukyu.ac.jp (R.T.); junkoba@med.u-ryukyu.ac.jp (J.K.); 3JICA Okinawa, Okinawa 901-2552, Japan; 4Pulmonary Medicine, Yuuai Medical Center, Tomigusuku 901-0224, Okinawa, Japan; ysato@yuuai.or.jp (Y.S.); tmatsumoto@yuuai.or.jp (T.M.); 5Department of Emergency and Critical Care Medicine, Graduate School of Medicine, University of the Ryukyus, Okinawa 903-0215, Japan; takeume@med.u-ryukyu.ac.jp

**Keywords:** chronic respiratory disease, evacuation routes, tsunami shelter, 6-min walk test (6MWT), people with disabilities (PWDs)

## Abstract

An earthquake with a magnitude of 8 or 9 is predicted to occur near the Ryukyu Islands in Japan, for which the Okinawa Prefecture is preparing countermeasures. Evacuating people to a safe shelter within the tsunami arrival time is a crucial countermeasure. This study aims to understand the vulnerabilities of patients with chronic respiratory diseases in southern parts of Okinawa during a tsunami evacuation, thereby calculating evacuation distance of vulnerable patients and creating individual evacuation plans. Data for chronic respiratory patients obtained in July 2021 from the hospitals in Okinawa Prefecture include age, gender, diagnosis, residence, nearest tsunami shelter, oxygen flow at rest and walking, and maximum walking distance for 6 min based on a 6-min walk test. A quantum geographic information system was used for mapping the data. The survival potential of patients with chronic respiratory disease was evaluated by using a tsunami inundation depth of one meter and the distance within which an evacuation can be performed until the first tsunami wave reaches the nearest evacuation shelter. Results revealed a low survival potential for respiratory disease patients under the current tsunami evacuation plan. The study suggests creating an individual evacuation plan for vulnerable patients involving families and medical staff and then conducting a drill for improving the plan.

## 1. Introduction

Compared to that of other countries globally, Japan’s land is more prone to natural hazards. These include disasters such as volcanic eruptions, landslides, typhoons, floods, earthquakes, and tsunamis. This vulnerability can be attributed to natural conditions such as weather, topography, and geology. The country has historically been hit by major disasters.

Among the many disasters that regularly plague Japan, tsunamis constitute one of the greatest threats to human life. Island coastal areas have suffered some of the greatest damage. The December 2004 Sumatra earthquake (magnitude 9.2) killed 280,000 people along the Indian Ocean coast [1], and the March 2011 Tohoku-Pacific Ocean earthquake (Great East Japan Earthquake, magnitude 9.0) killed more than 18,000 people [2]. Even the waves that arrive on nearby shorelines only minutes after an earthquake occurrence in near-field tsunamis with an epicenter close to the land are significant threats to human life because they provide only short response periods for people to evacuate [3].

The island of Okinawa is in the southwest corner of the Japanese archipelago is surrounded by the East China Sea on the west side and the Pacific Ocean on the east side as shown in Figure 1a. Many residential areas and industries are concentrated in the lowlands along the coast below 5 m above sea level. Owing to these lowland residential conditions, 10,000 people were killed in the tsunami caused by the 24 April 1771, Yaeyama Earthquake. This is known as the Great Tsunami of the Meiwa era. More recently, as a result of the Chilean tsunami caused by the Chilean earthquake (Mw 9.5) in South America in 1960, a tsunami of 4 m was recorded on Okinawa Main Island, and several people died, even though the epicenter was far from Japan [4].

As Figure 1b shows, recent studies predict that there is a high possibility of a future large earthquake of magnitude 8 or 9 occurring within the epicenter of the Ryukyu Trench near the Ryukyu Islands [5]. In response to this earthquake prediction, the Okinawa Prefecture has been promoting tsunami countermeasures since 2015. These countermeasures include making tsunami inundation area predictions [6] and conducting tsunami and storm surge damage estimation surveys. However, it is difficult to prevent a huge tsunami with only breakwaters and other defense facilities, so it is important to evacuate to avoid casualties by “running to a safe hill or building” within the tsunami arrival time [7].

The Basic Act on Disaster Management was enacted in 1961 as an evacuation guideline for residents in the event of a disaster in Japan. The devastating tsunami damage caused to the Pacific coasts by the Tohoku Earthquake triggered a re-examination of the framework of the Basic Act on Disaster Countermeasures. The 2013 revision mentions elderly people and persons with disabilities who need special consideration for evacuation in the event of a disaster. This resulted in the creation of lists of people requiring support for evacuation action in about 99% of municipalities [8]. Unfortunately, many elderly people were also injured during the subsequent disaster, and the issue became how to ensure the effectiveness of evacuation. To deal with this issue, a part of the Basic Act on Disaster Countermeasures was revised in 2021, and it was pointed out that an individual evacuation plan for people requiring evacuation action support was necessary. However, there are only a limited number of studies on the evacuation of people who have special illnesses or physical disabilities. According to the Okinawa Prefecture Tsunami Evacuation Plan Formulation Guidelines [9], the standard evacuation speed is 1.0 m/s free walking speed for physically healthy adults and elderly people, but the average walking speed drops to 0.5 m/s for people with walking difficulties, physically handicapped people, infants, and seriously ill people. Patients with chronic respiratory disease who have dyspnea during a walk, who have decreased exercise tolerance, and who need to use medical equipment such as home oxygen therapy (HOT) should receive special consideration for evacuation behavior from the tsunami and their evacuation distance. Despite the need for individualized evacuation plans, the response is considered inadequate by a medical staff and patients’ families. 

The aim of this study is to calculate the evacuation distance using the walking speed based (m/s) on the results of a 6-min walk test (6MWT) [10] of patients with chronic respiratory disease with HOT who live in the southern areas of the main island in Okinawa Prefecture, Japan and to clarify their possibility for survival based on the estimated tsunami time. Using a geographical information system (GIS) [11], we also identified the nearest evacuation shelters and places for patients with chronic respiratory disease. We believe that the results of this study will clarify the vulnerabilities to tsunamis of patients with chronic respiratory diseases and contribute to disaster countermeasures for families and visiting nurses who support evacuation behavior, and to the creation and training of individual evacuation plans

## 2. Materials and Methods

### 2.1. Participants and Data Collection

We conducted this study with the cooperation of two hospitals and their patients in the southern region of the main island of Okinawa Prefecture, Japan. These hospitals treat patients with chronic respiratory diseases who require treatment by a pulmonologist. The participants of the study are patients with chronic respiratory diseases with diagnoses such as chronic obstructive pulmonary disease (COPD) and interstitial pneumonia (IP). Hospital medical records obtained in July 2021 revealed age, gender, diagnoses, residential addresses of patients, the address of the nearest tsunami shelter (tsunami evacuation building), type of HOT, oxygen flow (L/min) at rest and walking, and maximum walking distance for 6 min based on the 6MWT. Walking speed (m/min) was calculated from the result of the 6MWT. For the tsunami first wave arrival time, the values shown in the Okinawa Prefecture Tsunami Evacuation Plan Formulation Guidelines [9] were used. 

### 2.2. Target Area

As Figure 1c shows, the target area was the southern part of the main island in Okinawa Prefecture. There are seven cities and towns in this area: Naha, Tomigusuku, Itoman, Yaese, Nanjyou, Yonabaru, and Haebaru. This area has a population of about 540,000. It faces the East China Sea to the west and the Pacific Ocean to the east. This is also an area that is expected to be damaged by a large tsunami when a magnitude 8 or 9 class submarine earthquake with an epicenter in the Ryukyu Trench occurs [5].

### 2.3. Mapping Locations of Chronic Respiratory Disease Patient and Tsunami Evacuation Area/Shelters

We obtained the addresses of patients with chronic respiratory diseases in the southern area of the main island in Okinawa Prefecture from the medical records of two hospitals. Information about tsunami evacuation areas and shelters in the study area were collected from the Okinawa Prefecture Tsunami Evacuation Plan Formulation Guidelines on the Okinawa Prefecture public website [9] and the designated emergency evacuation shelter map (tsunami) on the Technical Report of the Geospatial Information Authority of Japan website [12], and their addresses were obtained from information published on the internet. Then, the addresses of patients’ homes and evacuation areas and shelters were converted into the latitudinal and longitudinal position data, using GIS. On all maps, the locations of the homes of patients with chronic respiratory diseases are shown as circles, and the evacuation shelters are shown as squares. We obtained information about the estimated tsunami inundation area from the Okinawa Prefecture website [9], which details the inundation area and depth for the largest class of tsunamis.

Mapping was performed using a quantum geographic information system 3.16.11. (https://qgis.org/ja/site/, accessed on 6 April 2022). Based on the Okinawa Prefecture Tsunami Evacuation Plan Formulation Guidelines, the relationship between the tsunami inundation depth and the tsunami damage is as follows:Green area (0.01 m or more and less than 0.3 m): Although the tsunami has reached the area, no human or building damage is expected.Yellow area (0.3 m or more and less than 1.0 m): Evacuation behavior is not possible.Orange area (1.0 m or more and less than 2.0 m): Most people die when caught in a tsunami.Pink area (2.0 m or more and less than 5.0 m): 50% of wooden houses will be destroyed.Purple area (5.0 m or more and less than 10 m): A two-story building is submerged.Red area (10 m or more and less than 20 m): A three-story building is completely submerged.

### 2.4. Analysis of Data on Patients with Chronic Respiratory Disease

We counted the number of patients with chronic respiratory diseases located in the predicted tsunami inundation area. The distance from the patient’s home to the respective nearest evacuation area and shelter was initially calculated using network analysis. Open Street Map was used to obtain road network data for the analysis. Google map distance measurement was also used to calculate the distance for the area where the road network data was not sufficiently up-to-date in Open Street Map.

Evacuation distance was calculated according to the following formula:*Ed* = *Ws* ∗ *Tunami Rt*
where *Ed* is the distance that can be evacuated until the tsunami first wave (m), *Ws* is the walking speed (m/min), *Tsunami Rt* is the tsunami reaching time (min).

The survival potential of patients with chronic respiratory diseases located in the tsunami inundation area was evaluated in accordance with the following two criteria: (1) The tsunami inundation depth is less than 1 m, and (2) the distance that can be evacuated until the tsunami first wave exceeds the distance to the nearest evacuation shelter.

## 3. Results

### 3.1. Participant Characteristics

Fifty-five patients with chronic respiratory diseases were included in this study. Table 1 shows the participants’ characteristics. The average age of the participants was 71.4 ± 15.7 years. There were 40 men and 15 women. The average distance to the nearest evacuation shelter was 428 ± 269 m, and the median was 400 m. The average of the 6MWT results was 97 ± 131 m, and the median was 50 m. The types of HOT were 24 for the liquid oxygen type and 31 for the oxygen concentrator type. As for therapeutic oxygen flow, the average resting oxygen flow was 1.38 ± 0.9 L/min, and the median was 1 L/min. However, during exercise, the average oxygen flow was 2.42 ± 1.36 L/min, and the median was 2 L/min.

### 3.2. Data on Patients with Chronic Respiratory Diseases in Tsunami Inundation Areas

Of the 55 patients with chronic respiratory diseases, 20 were located in the tsunami inundation areas. Table 2 shows data on patients with chronic respiratory illness in these areas. The average age of these 20 patients was 67.5 ± 19.8 years. The gender distribution was 14 males and 6 females. Patient diagnoses included IP, bronchiectasis (BE), COPD, pulmonary artery hypertension (PAH), combined pulmonary fibrosis and emphysema (CPFE), Shintzen–Goldberg syndrome (SGS), and pulmonary tuberculosis sequelae (PTS).

The average of walking speed calculated from the 6MWT was 15 ± 22 m (median 7 m, 0–92 m). The average of the tsunami first wave arrival time was 16 ± 5 min (median 15 min, 6–23 min). The average of the evacuation distance was 248 ± 367 m (median 97.5 m, 0–1380 m). The average distance to shelter was 458 ± 312 m (median 400 m, 7–1200 m). Twelve patients were predicted to be able to survive the tsunami, and eight were predicted to have difficulty surviving the tsunami.

### 3.3. Locations of Patients with Chronic Respiratory Diseases

Figure 2 shows the locations of patients with chronic respiratory diseases with HOT who live in the southern part of the main island of Okinawa used in this study. The tsunami inundation area is colored in light blue. Twenty patients from the two hospitals that participated in this study lived in this tsunami inundation area. There are a few areas in which patients will be challenged to survive owing to tsunami predictions (Figure 2). One area is the western coastline on the southern part of areas of the main island of Okinawa (Figure 3 and Figure 4). The other is the coastline containing an islet on the southeast side (Figure 5). Figure 3 shows one of the tsunami inundation areas on the western coast of this southern Okinawa region. This area includes parts of Tomigusuku and Itoman cities. It is estimated that five patients with chronic respiratory diseases (one orange circle, two green circles, one light green circle, and one light blue circle) will have difficulty surviving the tsunami. In addition, the evacuation distance was not a problem for the patients indicated by the lowercase letter “a”, but it was necessary to pass over the estuary and bridge on the evacuation route. Figure 4 shows another of the tsunami inundation areas on the western coast of southern Okinawa. This area includes a part of Itoman City. It is estimated that one patient with chronic respiratory diseases (patient “b”) will be affected by the tsunami in this area. Since the evacuation site for the patient is located outside the tsunami inundation area, it is possible for the patient to survive if he or she can evacuate to the outside the tsunami inundation area by the time the first wave of the tsunami arrives. However, the patient’s self-evacuation distance—the maximum distance of self-mobility within the time that the first tsunami wave reached in—is 20 m only. Figure 5 shows the tsunami inundation area on the coastline containing an islet on the southeast side of the southern part of Okinawa. This area includes a part of Yaese and Nanjyo cities. It is estimated that two patients with chronic respiratory diseases will be affected by the tsunami in this area. Since the evacuation sites for two patients are located on the inland side of the tsunami inundation area, it is possible for the patients to survive if he or she can evacuate to the outside of the tsunami inundation area by the time the first tsunami arrives. However, the distance that the patient with lowercase “c” can evacuate on their own is only 49 m. On the contrary, patient with a lowercase “d” cannot evacuate him- or herself because of chronic respiratory illness.

## 4. Discussion

The current tsunami evacuation plan in Okinawa Prefecture recommends that evacuation plans be formulated by classifying evacuation speeds according to distinct physical profiles, such as physically healthy, elderly, or differently abled people. However, since patients with chronic respiratory diseases are considered to have individual differences in evacuation speed depending on the type and severity of their diseases and types of HOT, in this study, the evacuation speed is calculated from the results of a 6MWT measured with the HOT device. Based on this calculation, we have predicted the possibility of survival from the tsunami disaster. Although there were a limited number of participants in this study, the results demonstrate that, with the current tsunami evacuation plan, there exist patients with chronic respiratory diseases who will not be able to escape to the evacuation shelter before the first tsunami wave reaches them.

Based on this study, we want to convey most strongly that for patients who do not meet these two criteria, it is imperative to create individualized evacuation plans. These individualized plans should include vertical evacuations and practical evacuation training based on individual needs, rather than a generalized evacuation plan that inadvertently excludes the differently abled [13,14,15]. To enable vertical evacuation, a building with sufficient height that can be used in the event of a disaster, such as a tsunami evacuation building, is required. Locating the evacuation building in the residential area of patients with chronic respiratory diseases, who are the subjects of this study, will be necessary to establish more inclusive evacuation plans capable of vertical evacuation in the tsunami inundation areas (Appendix A) [12]. Of the patients in this study, 24 used liquid oxygen-type HOT equipment, but liquid oxygen equipment is not suitable for evacuation behavior owing to the challenges of oxygen storage and replenishment. For patients with chronic respiratory diseases who use oxygen concentrator equipment at home, evacuating with a 400-L portable oxygen cylinder weighing 2.5 kg is not feasible. In addition, since the median oxygen flow rate during exertion in this study is 2 L/min, it is calculated that even with a 400-L oxygen cylinder, the remaining amount only suffices for 3 h when used without a respiratory synchronization device. Therefore, oxygen replenishment even after evacuation is a pressing issue [16]. It is known that the interruption of oxygen supply to HOT patients causes health hazards synergistically with the tsunami disaster [17,18].

We have experience from the 2011 off the Pacific coast of Tohoku Earthquake (Great East Japan Earthquake) with this problem [19]. However, Okinawa Prefecture is an island prefecture. Therefore, assistance in the event of damage must cross the sea. Although disaster countermeasures are the responsibility of local governments, problems related to HOT in the event of a disaster will need to be addressed at the national government level [20].

Furthermore, as in the example of the patient indicated by lowercase “a” in Figure 3, even if the tsunami inundation depth is not life-threatening, a separate evacuation plan is required if the evacuation route to the evacuation shelter passes over water or an estuary. In such cases, it will be necessary to reexamine the individual evacuation plan considering not only distance but also the risks posed by the evacuation routes themselves.

From the perspective of natural disaster prevention and risk reduction, there are several previous studies on tsunami damage [21,22,23]. These studies have demonstrated that it is particularly difficult to prevent a large tsunami with only breakwaters and other defensive measures. Therefore, it is imperative to evacuate by “running to a safe hill or building to prevent as many casualties as possible” within the tsunami arrival time.

Several previous studies on disaster countermeasures for vulnerable people such as people with disabilities and have been reported [11,24,25,26,27,28,29]. In addition, our tsunami evacuation maps using GIS have already been reported [30,31].

However, there is a dearth or complete absence of survey reports addressing the tsunami risk to home-care patients with chronic respiratory diseases, except for surveys on specific disaster countermeasures for patients with COPD and amyotrophic lateral sclerosis [32,33]. It is noteworthy that we applied the results of 6MWT to tsunami evacuation of patients with chronic respiratory diseases.

Based on the results of this research, there are some matters to be considered in the future. This study only predicted the vulnerabilities of patients with chronic respiratory diseases to tsunami damage from the estimated evacuation distance on a map. The true value of this study is that based on the results of this study, patients with chronic respiratory diseases, their families, communities, or home medical doctors and visiting nurses will practice individual evacuation plans and training tailored to each patient’s characteristics [34,35].

## 5. Limitations

The present study has several limitations. First, it was conducted with the cooperation of two hospitals only, including their patients in the southern part of Okinawa Prefecture. Therefore, the results are based on a limited number of patients. Second, the walking distance of the 6MWT in this study was measured in the presence of a physiotherapist in a flat, unobstructed corridor in a hospital. Therefore, there may be a difference between the distance in the actual tsunami evacuation and the estimated evacuation distance based on the 6MWT. Third, regarding evacuation routes, there may be slopes, stairs, and road closures, so it is not possible to identify the optimal evacuation route using the map alone. In addition, in the event of a tsunami disaster, there is a possibility that the shortest evacuation route from one’s home to the tsunami shelter, which was used in the estimation on the map this time, may not be passable owing to fires or building collapse. Fourth, the formula for calculating the evacuation distance used in this study does not consider the time for each patient with a chronic respiratory disease to decide to start evacuation behavior. However, it has been reported that various factors affect the time it takes for people to start evacuation behavior after the tsunami warning is issued [1,36,37,38]. Therefore, it is believed that there is a time lag from the issuance of the tsunami warning to the start of evacuation behavior, and the actual evacuation distance may be shorter than the distance estimated in this study. This perspective should also be considered in individual evacuation plans.

## 6. Conclusions

Based on the evacuation speed calculated from the results of a 6MWT measured using the HOT device, we predicted the possibility of survival of patients with chronic respiratory disease is a tsunami disaster. The results showed that, with the current tsunami evacuation plan, there are patients with chronic respiratory disease who cannot escape to an evacuation shelter by the time the first tsunami wave arrives. The true value of this study is that based on its results, patients with chronic respiratory diseases, their families, communities, or home medical doctors and visiting nurses will practise individual evacuation plans such as vertical evacuation and training tailored to each patient’s circumstances.

## Figures and Tables

**Figure 1 ijerph-19-05647-f001:**
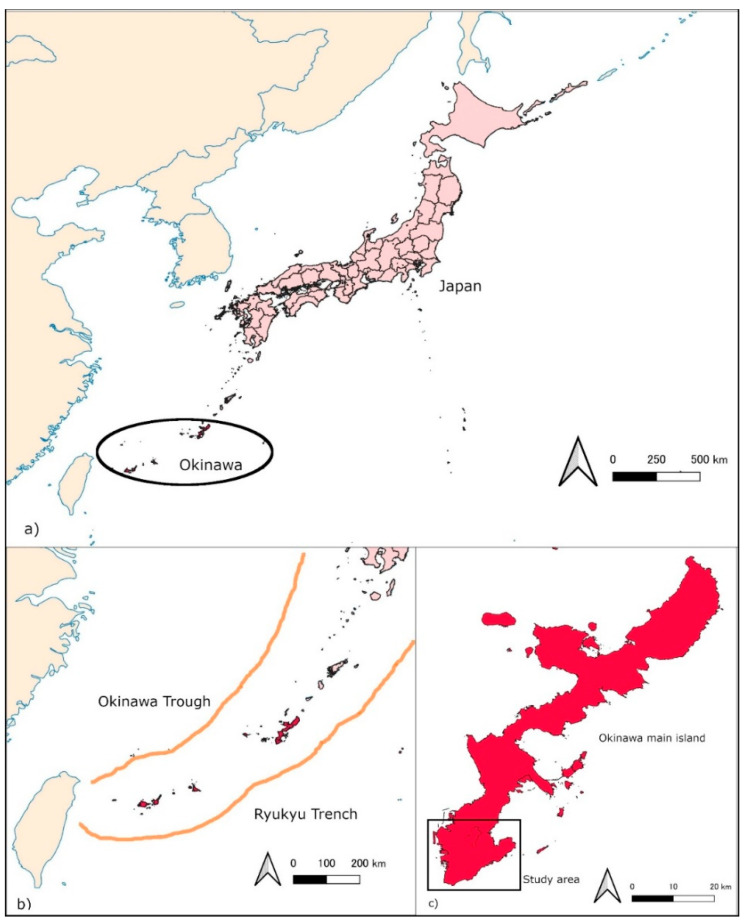
(**a**): locations of the Okinawa Prefecture; (**b**): the Ryukyu Trench; (**c**): the southern area of the main island in Okinawa Prefecture.

**Figure 2 ijerph-19-05647-f002:**
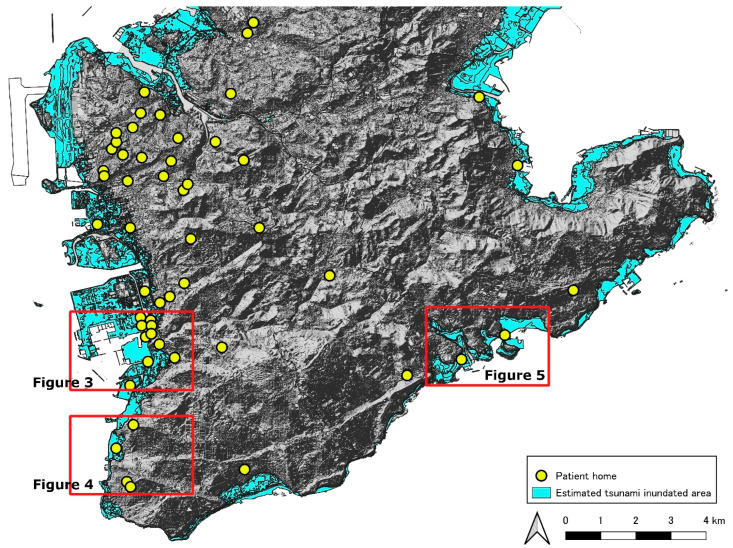
The locations of patients with chronic respiratory diseases with HOT who live in the southern areas of the main island of Okinawa.

**Figure 3 ijerph-19-05647-f003:**
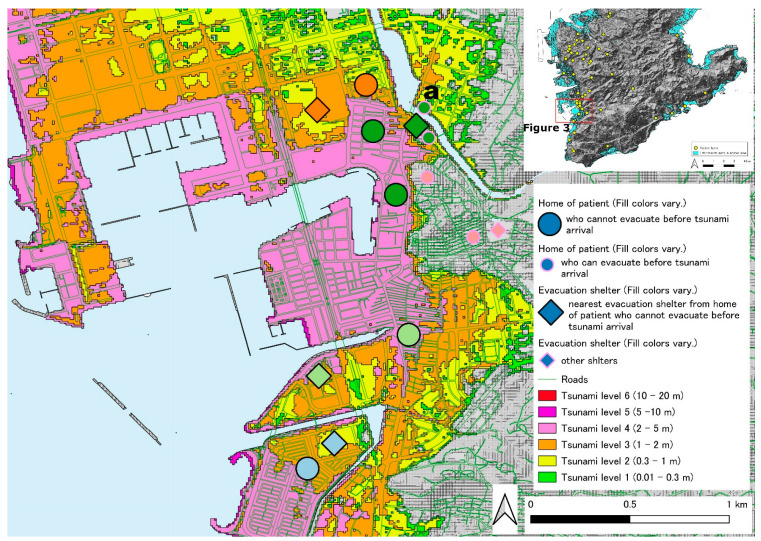
The locations of patients with chronic respiratory diseases with HOT who live in the tsunami inundation areas on the western coast of the southern part of the main island of Okinawa-1.

**Figure 4 ijerph-19-05647-f004:**
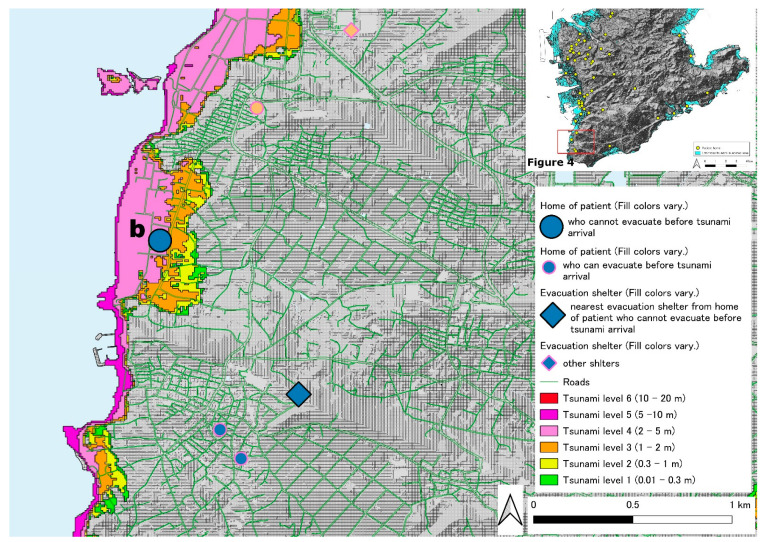
The locations of patients with chronic respiratory disease with HOT who live in the tsunami inundation areas on the western coast of the southern part of the main island of Okinawa-2.

**Figure 5 ijerph-19-05647-f005:**
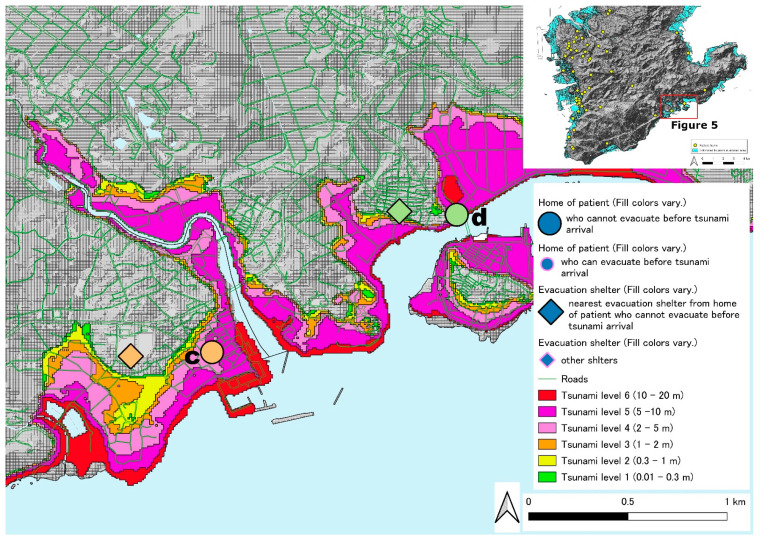
The locations of patients with chronic respiratory diseases with HOT who live near the coastline containing an islet on the southeast side of the southern part of the main island of Okinawa.

**Table 1 ijerph-19-05647-t001:** The characteristics of patients (*n* = 55).

Item	Category	Number
**Age**	Average ± standard deviation	71.4 ± 15.7 (year)
**Gender**	Male	40
	Female	15
**Nearest shelter**	Average ± standard deviation	428 ± 269
	Median	400
**6 min walk test**	Average ± standard deviation	97 ± 131
	Median	50
**Types of HOT**	Liquid Oxygen	24
	Oxygen concentrator	31
**Oxygen flow on rest**	Average ± standard deviation	1.38 ± 0.9
	Median	1
**Oxygen flow on exercise**	Average ± standard deviation	2.42 ± 1.36
	Median	2

**Table 2 ijerph-19-05647-t002:** Data on patients with chronic respiratory diseases in tsunami inundation.

Number	Age	Gender	Diagnosis	Tsunami ID (m)	Ws (m/min)	Tsunami First Wave Rt (min)	Ed (m)	Distance to Shelter (m)	Survival (1-0)
1	57	M	IP	0.3–1.0	40	23	920	450	1
2	83	M	BE	2.0–5.0	43	19	817	350	1
3	67	F	IP	1.0–2.0	92	15	1380	400	1
4	27	M	PAH	2.0–5.0	8	15	120	450	0
5	75	F	BE	2.0–5.0	7	22	154	7	1
6	65	M	IP	0.01–0.3	13	19	247	1000	1
7	88	F	COPD	0.01–0.3	17	22	374	500	1
8	73	F	IP	0.3–1.0	0	21	0	400	1
9	85	M	COPD	2.0–5.0	2	15	30	550	0
10	83	M	COPD	0.3–1.0	17	10	170	1100	1
11	63	M	COPD	5.0–10.0	0	6	0	400	0
12	85	F	BE	2.0–5.0	2	10	20	1200	0
13	74	M	IP	2.0–5.0	2.5	15	38	270	0
14	68	M	COPD	10.0–20.0	7	7	49	400	0
15	91	M	CPFE	1.0–2.0	5	15	75	66	1
16	27	F	SGS	0.01–0.3	5	15	75	230	1
17	32	M	SGS	0.01–0.3	7	15	75	230	1
18	71	M	PTS	0.01–0.3	2	22	44	300	1
19	51	M	CPFE	2.0–5.0	17	15	255	550	0
20	85	M	COPD	2.0–5.0	8	15	120	300	0

Abbreviations: IP, interstitial pneumonia; BE, bronchiectasis; COPD, chronic obstructive pulmonary disease; PAH, pulmonary artery hypertension; CPFE, combined pulmonary fibrosis and emphysema; SGS, Shprintzen–Goldberg syndrome; PTS, pulmonary tuberculosis sequelae; Tid, tsunami inundation depth; Ws, walking speed; tsunami first wave Rt, tsunami first wave reach time; Ed, evacuation distance. Survival means the survival potential of patients with chronic respiratory disease. Number 1 means that there is a possibility of survival. Number 0 indicates that it is difficult to survive.

## Data Availability

Not applicable.

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
