# Peer review of "Can Homecare Chronic Respiratory Disease Patients with Home Oxygen Treatment (HOT) in Southern Okinawa, Japan Be Evacuated Ahead of the Next Anticipated Tsunami?"

_ijerph, 2022, doi:10.3390/ijerph19095647_

Round 1
Reviewer 1 Report
Dear Authors,
The introduction of the manuscript provides sufficient information and includes all relevant links.
The research design is good, the methods are well described.
The results are clearly presented.
However, I believe that the aim of the research and its results are not new. Several researchers have already dealt with a similar topic. The studies have a similar result. For example:
Dall'Osso, F. and Dominey-Howes, D. (2010). Public assessment of the usefulness of "draft" tsunami evacuation maps from Sydney, Australia – implications for the establishment of formal evacuation plans, Nat. Hazards Earth Syst. Sci., 10, 1739–1750, https://doi.org/10.5194/nhess-10-1739-2010
Nakai, H., Itatani, T., Horiike, R., Kyota, K., & Tsukasaki, K. (2018). Tsunami evacuation simulation using geographic information systems for homecare recipients depending on electric devices. PloS one, 13(6), e0199252. https://doi.org/10.1371/journal.pone.0199252
However, your manuscript is interesting to readers.
Kind regards.
Author Response
Response to reviewer 1.
Reviewer 1.
Dear Authors,
The introduction of the manuscript provides sufficient information and includes all relevant links.
The research design is good, the methods are well described.
The results are clearly presented.
However, I believe that the aim of the research and its results are not new. Several researchers have already dealt with a similar topic. The studies have a similar result. For example:
Dall'Osso, F. and Dominey-Howes, D. (2010). Public assessment of the usefulness of "draft" tsunami evacuation maps from Sydney, Australia – implications for the establishment of formal evacuation plans, Nat. Hazards Earth Syst. Sci., 10, 1739–1750, https://doi.org/10.5194/nhess-10-1739-2010
Nakai, H., Itatani, T., Horiike, R., Kyota, K., & Tsukasaki, K. (2018). Tsunami evacuation simulation using geographic information systems for homecare recipients depending on electric devices. PloS one, 13(6), e0199252. https://doi.org/10.1371/journal.pone.0199252
However, your manuscript is interesting to readers.
Kind regards.
Response.
We appreciate your valuable comment. As you pointed out, previous studies on disaster countermeasures using GIS have been reported. However, we believe that the combination of the evacuation distance based on the result of 6MWT and the map by GIS is our originality. The references you have introduced have been added to the discussion “In addition, our tsunami evacuation maps using GIS have already been reported [30][31] (line 289).”
Moreover, we modified lines 293–294 as follows: “It is noteworthy that we applied the results of 6MWT to tsunami evacuation of patients with chronic respiratory diseases.”

Reviewer 2 Report
I thank the authors for letting me read this interesting manuscript.
In order to improve the manuscript, I allow myself to make some observations.
The title can be misleading because it does not clearly indicate that patients with chronic respiratory diseases who use HOT are being studied. This information (HOT) should be included in the title.
In the materials and methods section, it is not clear to me if all the hospitals in the area studied are included or just two. In other countries, HOT is not only prescribed in the hospital but also outside the hospital (for example, by family doctors). In Okinawa, is HOT only prescribed in hospitals? Do hospitals have a registry of all patients with HOT prescribed outside hospitals? Have these circumstances been contemplated in this study?
I believe that the information contained in the paragraphs of text lines 189-201 can be implemented in table 2 and these paragraphs deleted. The information of the first paragraph can be added in a new column and the information of the second paragraph can be implemented in one more row.
Rows 235-236, 240-244, 248-252 and 256-260 must be eliminated since this information is already included in the figures.
The second paragraph of the discussion should be deleted as it is repetitive and has been written literally in the materials and methods section.
Author Response
The title can be misleading because it does not clearly indicate that patients with chronic respiratory diseases who use HOT are being studied. This information (HOT) should be included in the title.
Response.
We appreciate your valuable advice. Following your advice, we added the term HOT to the title.
In the materials and methods section, it is not clear to me if all the hospitals in the area studied are included or just two. In other countries, HOT is not only prescribed in the hospital but also outside the hospital (for example, by family doctors). In Okinawa, is HOT only prescribed in hospitals? Do hospitals have a registry of all patients with HOT prescribed outside hospitals? Have these circumstances been contemplated in this study?
Response.
This point was described in the Participants and Data Collection section. However, we apologize for the inadequate explanation. The description here has been revised as follows.
“We conducted this study with the cooperation of two hospitals and their patients in the southern region of the main island of Okinawa Prefecture, Japan. (lines 104-105)”
Only two hospitals participated in this study, and patients are limited to those hospitals. In Okinawa, HOT prescriptions are introduced not only in hospitals but also in clinics. There is probably no formal and unified list of all HOT patients and to recognize the location of wider HOT patients will require the cooperation of oxygen supply companies. We mentioned that the subjects of this study were a limited number of HOT patients in Okinawa in the Limitations section.
I believe that the information contained in the paragraphs of text lines 189-201 can be implemented in table 2 and these paragraphs deleted. The information of the first paragraph can be added in a new column and the information of the second paragraph can be implemented in one more row.
Response.
Thank you for your recommendations. The information in the first paragraph of lines 189–201 has been implemented in column 5 of table 2 from the time this manuscript was created. Therefore, according to your advice, we deleted the paragraphs from lines 189–195. Since the content of the next paragraph (lines 191–196) is based on the data in Table 2, we thought it difficult to implement it again in Table 2. Thank you for the understanding.
Rows 235-236, 240-244, 248-252 and 256-260 must be eliminated since this information is already included in the figures.
Response.
We appreciate your recommendations. We considered your comments for the data included in each figure. As a result, the descriptions in lines 235–236, 240–244, 248–252, and 256–260 have been deleted.
The second paragraph of the discussion should be deleted as it is repetitive and has been written literally in the materials and methods section.
Response.
Thank you for your valuable comments. We have deleted the second paragraph of the discussion based on your advice.

Reviewer 3 Report
General comment
This is a multi-institutional collaborative study to evaluate the possibility of survival if patients with chronic respiratory diseases treated by home oxygen treatment (HOT) tried to escape from the tsunami by a possible M8-9 earthquake at Ryukyu trench. I truly appreciate the voluntary contribution of the patients and their healthcare providers to assess the individual evacuation plan based on the expected tsunami inundation, arrival time and the coping capacity of patients measured by 6MWT. It will definitely give the patients to understand the difficulty for them to evacuate to the shelter horizontally and to prepare the individual evacuation plan with families, and healthcare providers to decide to stay in their own place using vertical evacuation or to escape horizontally.
I strongly recommend describing the type of this study in the abstract and introduction because this study seems to be a prospective study to ask patients to participate voluntarily in giving their personal information such as the location of residence, age, gender, and 6MWT with HOT devices. I wonder whether you have obtained other information including the type of house structure (wooden, reinforced concrete which is very popular in Okinawa), whether the house has upper stories that will help the vertical evacuation, or they have their willingness to evacuate horizontally or vertically, or they have their own or family plan for evacuation, whether they have extra cylinders or batteries for emergency situation.
How do you plan to return (or have returned) the results of this study to patients and their caregivers?
As you mention, a liquid oxygen cylinder is not enough to support a long evacuation time. Do you have a specific plan for the health facilities to provide additional cylinders to these patients?
There are at least four English articles in PubMed using “great east japan earthquake, home oxygen treatment”, which could be cited in this article.
1: Kobayashi S, Hanagama M, Yamanda S, Yanai M. Home oxygen therapy during natural disasters: lessons from the great East Japan earthquake. Eur Respir J. 2012 Apr;39(4):1047-8. doi: 10.1183/09031936.00149111. PMID: 22467730.
2: Kobayashi S, Hanagama M, Yamanda S, Satoh H, Tokuda S, Kobayashi M, Ueda S, Suzuki S, Yanai M. Impact of a large-scale natural disaster on patients with chronic obstructive pulmonary disease: the aftermath of the 2011 Great East Japan Earthquake. Respir Investig. 2013 Mar;51(1):17-23. doi:10.1016/j.resinv.2012.10.004. Epub 2012 Dec 27. PMID: 23561254.
3: Sato K, Morita R, Tsukamoto K, Sato N, Sasaki Y, Asano M, Okuda Y, Miura H, Sano M, Kosaka T, Watanabe H, Shioya T, Ito H. Questionnaire survey on the continuity of home oxygen therapy after a disaster with power outages. Respir Investig. 2013 Mar;51(1):9-16. doi: 10.1016/j.resinv.2012.10.005. Epub 2012 Dec 27. PMID: 23561253.
4: Kida K, Motegi T, Ishii T, Hattori K. Long-term oxygen therapy in Japan: history, present status, and current problems. Pneumonol Alergol Pol. 2013;81(5):468-78. PMID: 23996887.
Minor points
- Line 18: This sentence lacks a verb.
- Line 21: Indicate that this study is a prospective study with patients' informed consent including measurement of 6MWT.
- LIne 23: Spell out 6MWT.
- Line 35: Please be careful on the difference of hazards and disasters. It is correct to say Japan is prone to "natural hazards". Disaster is not natural because it always associates human society. Refer to agreed terminology on disaster at https://www.undrr.org/terminology
- Line 40: Use "disasters" instead of "natural disasters".
- Line 93: HOT is already spelled out before.
- Line 136: Add the website URL of the QGIS system (https://qgis.org/ja/site/).
- Line 215: Because you have obtained information from all patients, I suppose that you might have got information on the possibility of vertical evacuation. Do you have data on whether a patient is living on the first floor or the possibility to evacuate to the nearest second or higher floor? It is also likely that Okinawan people are living in reinforced concrete (RC) houses to survive frequent typhoon attacks. Do you have data on the housing structure of these patients? RC houses may endure the Orange (1-2m) inundation.
- Line 219: "for this patient (b)" makes it easier to understand the specific situation.
- Line 222: "self-evacuation distance" appeared for the first time and used only once. Add an explanation whether it means the maximum distance of self mobility within a certain time (6 min, an hour, or what?).
- Line 226: "inland side" is better to explain the location of this patient b than "outside".
- Line 229: Lower case "d" indicates the remaining one patient. Correct "patients" and "themselves".
- Line 311: I strongly agree with your originality to ask patients voluntarily participate in this study to provide the locatiion of their residency and the measurement of 6MWT. You should stress more about this study profile in the abstract, introduction, and patients and methods sections.
- Line 342: "escape to survival in an evacuation center" sounds strange. Simply, "escape to an evacuation center" may fit because it is not also assured that an evacuation is the safest place to survive especially for these patients.
- Line 364: It is not clear whether the authors have asked patients to give written informed consent, or not. Please clarify. If you did not get informed consent, "participants" sounds strange. If you obtained written informed consent, please clarify how many of the patients were asked to participate, and how many accepted to participate.
Author Response
Response to reviewer 3.
Reviewer 3.
General comment
This is a multi-institutional collaborative study to evaluate the possibility of survival if patients with chronic respiratory diseases treated by home oxygen treatment (HOT) tried to escape from the tsunami by a possible M8-9 earthquake at Ryukyu trench. I truly appreciate the voluntary contribution of the patients and their healthcare providers to assess the individual evacuation plan based on the expected tsunami inundation, arrival time and the coping capacity of patients measured by 6MWT. It will definitely give the patients to understand the difficulty for them to evacuate to the shelter horizontally and to prepare the individual evacuation plan with families, and healthcare providers to decide to stay in their own place using vertical evacuation or to escape horizontally.
I strongly recommend describing the type of this study in the abstract and introduction because this study seems to be a prospective study to ask patients to participate voluntarily in giving their personal information such as the location of residence, age, gender, and 6MWT with HOT devices. I wonder whether you have obtained other information including the type of house structure (wooden, reinforced concrete which is very popular in Okinawa), whether the house has upper stories that will help the vertical evacuation, or they have their willingness to evacuate horizontally or vertically, or they have their own or family plan for evacuation, whether they have extra cylinders or batteries for emergency situation.
Response.
We appreciate your detailed review of our manuscript and your valuable comments.
First, we apologize for the inadequate explanation of our study. 6MWT was not necessarily newly measured for this study, and the latest data measured at outpatient or inpatient wards were used. The data was acquired retrospectively from the medical record. From that point of view, the Research Ethics Review Committee considered that no new physical intervention was placed on the patient for this study.
After informing patients of the existence and purpose of the study in the outpatients’ ward, we emphasized that participation was voluntary, that refusing to cooperate would not affect them detrimentally, and that they could withdraw from participation at any time.
Unfortunately, we do not think this study can be a prospective study to ask patients to participate voluntarily in giving their personal information, such as the location of residence, age, gender, and 6MWT with HOT devices. The “Informed Consent Statement” part in the manuscript has been added and revised. At the next stage of this study, interventions (individual evacuation drills) will exist. Therefore, we are going to design a prospective study based on your advice.
Following this study, we decided to proceed with gathering information on the specific structure of each home, the possibility of vertical evacuation and physical activities of vertical evacuation, alternative evacuation sites around the home, and individual evacuation drills. We had planned to coordinate with doctors, visiting nurses, physical therapist, and certified nurses in chronic respiratory nursing. However, due to the impact of the COVID-19 epidemic, individual investigations have stopped, and we have not made significant advancement. When the threat of COVID-19 abates, we plan to continue investigating this plan.
How do you plan to return (or have returned) the results of this study to patients and their caregivers?
Response.
Based on the results of this study, doctors, visiting nurses, physical therapists, and certified nurses in chronic respiratory nursing planned to work together to develop an individual evacuation plan. Some of the patients of this study were introduced by the attending physician to an individual evacuation plan based on the results of this study. Moreover, specific house surveys and evacuation drill consultations were being promoted. As mentioned in the response above, due to the impact of COVID-19 epidemic, individual investigations have stopped, and we have not made significant advancement. When the threat of COVID-19 abates, we plan to continue to investigate this plan.
As you mention, a liquid oxygen cylinder is not enough to support a long evacuation time. Do you have a specific plan for the health facilities to provide additional cylinders to these patients?
Response.
In a single facility, such as a hospital, the inpatient oxygen supply is prioritized, and HOT patients may not receive sufficient oxygen. In fact, in this COVID-19 epidemic, an oxygen supply problem in Okinawa Prefecture approached, and the oxygen supply provider promoted the stockpiling of oxygen concentrators as a business continuity plan against the epidemic, and the collapse of medical care in Okinawa was avoided. However, it cannot be denied that the tsunami disaster may be a more serious crisis. This is a serious issue facing Okinawa, an island prefecture.
There are at least four English articles in PubMed using “great east japan earthquake, home oxygen treatment”, which could be cited in this article.
1: Kobayashi S, Hanagama M, Yamanda S, Yanai M. Home oxygen therapy during natural disasters: lessons from the great East Japan earthquake. Eur Respir J. 2012 Apr;39(4):1047-8. doi: 10.1183/09031936.00149111. PMID: 22467730.
2: Kobayashi S, Hanagama M, Yamanda S, Satoh H, Tokuda S, Kobayashi M, Ueda S, Suzuki S, Yanai M. Impact of a large-scale natural disaster on patients with chronic obstructive pulmonary disease: the aftermath of the 2011 Great East Japan Earthquake. Respir Investig. 2013 Mar;51(1):17-23. doi:10.1016/j.resinv.2012.10.004. Epub 2012 Dec 27. PMID: 23561254.
3: Sato K, Morita R, Tsukamoto K, Sato N, Sasaki Y, Asano M, Okuda Y, Miura H, Sano M, Kosaka T, Watanabe H, Shioya T, Ito H. Questionnaire survey on the continuity of home oxygen therapy after a disaster with power outages. Respir Investig. 2013 Mar;51(1):9-16. doi: 10.1016/j.resinv.2012.10.005. Epub 2012 Dec 27. PMID: 23561253.
4: Kida K, Motegi T, Ishii T, Hattori K. Long-term oxygen therapy in Japan: history, present status, and current problems. Pneumonol Alergol Pol. 2013;81(5):468-78. PMID: 23996887.
Response.
We appreciate your comments. We considered the content of the literature and cited it as [17]–[20].
Minor points
- Line 18: This sentence lacks a verb.
Response.
We have confirmed and revised it.
- Line 21: Indicate that this study is a prospective study with patients' informed consent including measurement of 6MWT.
Response.
As described in the above response, the 6MWT of this research was not necessarily newly measured for this study, and the latest data measured at an outpatient or inpatient ward were used. The data was acquired retrospectively from the medical record. The Research Ethics Review Committee considered this to be no new physical intervention placed on the patient for this study. After all, this study is not a prospective study but rather a retrospective study.
- Line 23: Spell out 6MWT.
Response.
We revised it according to your advice.
- Line 35: Please be careful on the difference of hazards and disasters. It is correct to say Japan is prone to "natural hazards". Disaster is not natural because it always associates human society. Refer to agreed terminology on disaster at https://www.undrr.org/terminology
Response.
We have revised the term according to your advice.
- Line 40: Use "disasters" instead of "natural disasters".
Response.
We have revised the term according to your advice.
- Line 93: HOT is already spelled out before.
Response.
We have revised it according to your advice.
- Line 136: Add the website URL of the QGIS system (https://qgis.org/ja/site/).
Response.
We have added the website according to your valuable advice.
- Line 215: Because you have obtained information from all patients, I suppose that you might have got information on the possibility of vertical evacuation. Do you have data on whether a patient is living on the first floor or the possibility to evacuate to the nearest second or higher floor? It is also likely that Okinawan people are living in reinforced concrete (RC) houses to survive frequent typhoon attacks. Do you have data on the housing structure of these patients? RC houses may endure the Orange (1-2m) inundation.
Response.
Thank you for your valuable comments. This research is a so-called two-dimensional study on a map based on the address of each patient. The attending doctors have shared that the homes of a few patients are two-storied. Home-visit nurses also have useful information. However, the systematic investigation of the possibility of vertical evacuation using the home is incomplete. There is a lack of information on the actual structure of the house, the availability of vertical evacuation, individual evacuation plans, and training. This will be addressed in the next study.
- Line 219: "for this patient (b)" makes it easier to understand the specific situation.
Response.
We have revised it based on your suggestive advice.
- Line 222: "self-evacuation distance" appeared for the first time and used only once. Add an explanation whether it means the maximum distance of self mobility within a certain time (6 min, an hour, or what?).
Response.
Thank you for your important comments. This “self-evacuation distance” was expressed as “the patient’s self-evacuation distance—the maximum distance of self-mobility within the time that the first tsunami wave reached in—...”.
- Line 226: "inland side" is better to explain the location of this patient b than "outside".
Response.
Thank you for your precise advice. We have revised it according to your suggestion.
- Line 229: Lower case "d" indicates the remaining one patient. Correct "patients" and "themselves".
Response.
We have revised this sentence based on your recommendation.
- Line 311: I strongly agree with your originality to ask patients voluntarily participate in this study to provide the locatiion of their residency and the measurement of 6MWT. You should stress more about this study profile in the abstract, introduction, and patients and methods sections.
Response.
We apologize for the inadequate explanation of this study again. We would like to highlight that the study design of this research is unfortunately not a prospective study.
- Line 342: "escape to survival in an evacuation center" sounds strange. Simply, "escape to an evacuation center" may fit because it is not also assured that an evacuation is the safest place to survive especially for these patients.
Response.
Thank you for your valuable comments. We have revised it according to your advice.
- Line 364: It is not clear whether the authors have asked patients to give written informed consent, or not. Please clarify. If you did not get informed consent, "participants" sounds strange. If you obtained written informed consent, please clarify how many of the patients were asked to participate, and how many accepted to participate.
Response.
We apologize for the inadequate explanation of this study again. As mentioned in the response above, please understand that the study design of this research is unfortunately not a prospective study. Therefore, we did not have to ask patients to give written informed consent.
